# Poly(heptazine imide) ligand exchange enables remarkable low catalyst loadings in heterogeneous metallaphotocatalysis

Liuzhuang Xing [1,3], Qian Yang [1,3], Chen Zhu [2,3], Yilian Bai [1], Yurong Tang [1] ✉, Magnus Rueping [2] ✉ & Yunfei Cai [1] ✉

The development of heterogeneous metallaphotocatalysis is of great interest for sustainable organic synthesis. The rational design and controllable preparation of well-defined (site-isolated) metal/photo bifunctional solid catalysts to meet such goal remains a critical challenge. Herein, we demonstrate the incorporation of privileged homogeneous bipyridyl-based Ni-catalysts into highly ordered and crystalline potassium poly(heptazine imide) (K-PHI). A variety of PHI-supported cationic bipyridyl-based Ni-catalysts ($L_nNi$-PHI) have been prepared and fully characterized by various techniques including NMR, ICP-OES, XPS, HAADF-STEM and XAS. The $L_nNi$-PHI catalysts exhibit exceptional chemical stability and recyclability in diverse C−P, C−S, C−O and C−N cross-coupling reactions. The proximity and cooperativity effects in $L_nNi$-PHI significantly enhances the photo/Ni dual catalytic activity, thus resulting in low catalyst loadings and high turnover numbers.

Over the past decade, homogeneous nickel catalysis has become a powerful tool for organic synthesis[1,2]. Merging nickel catalysis with photoredox catalysis[3] enabled numerous challenging and valuable transformations. In this regard, a wide range of C−heteroatom and C−C cross-coupling reactions have been achieved, allowing for rapid access to privileged structure motifs prevalent in pharmaceuticals and functional materials (Fig. 1a)[4-19]. In these homogeneous Ni/photo dual catalytic systems, the highly reactive open-shell Ni(I)/Ni(III) or excited Ni(II) intermediates can be generated through photoinduced electron or energy transfer processes, allowing the transformations occur under very mild reaction conditions. Importantly, the choice of ligands plays a vital role in enhancing the nickel catalysts' reactivity, stability, and selectivity. Despite the elegance and versatility of the existing homogeneous methods, the development of heterogeneous metallaphotocatalysis by the rational design and construction of Ni-photo bifunctional catalysts, with the advantage of facile separation and catalyst reusability, is of great interest for industrial implementation and thus in urgent demand[20,21]. Of note, Ni complexes featuring bidentate bipyridyl-based ligands such as 2,2′-bipyridine (bpy), 4,4′-di-

tert-butyl-2,2′-bipyridine (dtbpy), and 1,10-phenanthroline (phen) have proven crucial for effective couplings in the homogenous Ni/photoredox dual catalysis[3,22]. These weak-field ligands lead to lower ligand field-splitting energy that promotes the formation of paramagnetic species and stabilization of open-shell Ni intermediates via metal-to-ligand charge transfer, thus significantly enhancing the reactivity of Ni center. Therefore, incorporating these privileged bipyridyl-based Ni catalysts into solid-state sensitizer materials holds great potential to furnish a novel heterogeneous platform for metallaphotoredox catalysis. The early research that employed carbon nitride ($C_3N_4$), perovskite, quantum dot (QDs) as heterogeneous photocatalyst in dual catalysis could only realize partial recycling due to the combined use of homogeneous Ni/ligand[23-29]. More recently, one approach has been developed to immobilize the engineered Ni complex containing carboxylic/phosphoric acid groups on the surface of dye-/carbon dot-sensitized titanium dioxide or carbon nitride photocatalyst[30,31]. Another attractive strategy was to incorporate Ni(II) and photosensitizing Ir centers into metal/covalent organic frameworks (MOFs/COFs)/flexible polymers containing chelating bpy/phen sites or

[1]School of Chemistry and Chemical Engineering, Chongqing University, 174 Shazheng Street, Chongqing 400044, P. R. China. [2]KAUST Catalysis Center (KCC), King Abdullah University of Science and Technology (KAUST), Thuwal 23955-6900, Saudi Arabia. [3]These authors contributed equally: Liuzhuang Xing, Qian Yang, Chen Zhu. ✉e-mail: tangyuronga@cqu.edu.cn; magnus.rueping@kaust.edu.sa; yf.cai@cqu.edu.cn

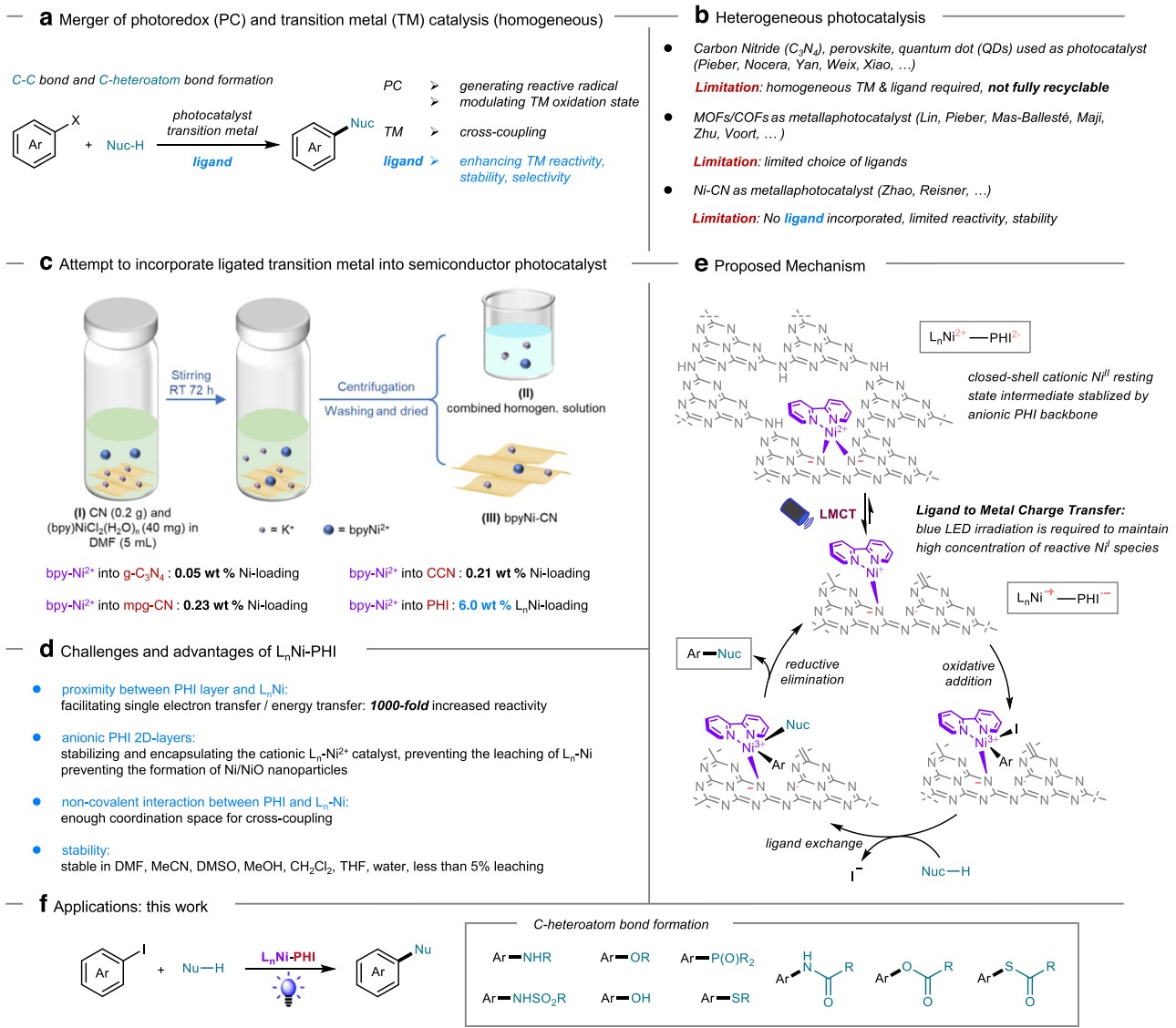

**Fig. 1 | Strategy for the ligand exchange and applications. a** Merger of homogeneous photoredox and transition metal catalysis. **b** Heterogeneous photocatalysis. **c** Attempt to incorporate ligated transition metal into semiconductor photocatalyst. **d** Challenges and advantages of $L_nNi$-PHI. **e** Proposed Mechanism. **f** Applications: this work.

directly embed Ni(II) in a photosensitive bpy-functionalized COF or poly-Czbpy[32–40]. Although appealing, these approaches may require multiple steps to access one solid catalyst with a specific bpy-based ligand, which often deleteriously affects the catalytic activity in comparison with the parent homogeneous Ni catalyst with tunable bpy-based ligands (Fig. 1b).

As a class of solid-state polymers, carbon nitrides have emerged as a promising light-harvesting material for applications in photocatalysis[41–50]. In addition, it is an appealing option for coordinating metals[51–53]. Recently, Zhao and Reisner developed integrated carbon nitride-nickel photocatalysts with exceptional recyclability in cross-coupling reactions[54–56]. However, these catalysts without bipyridyl-based ligand exhibited limited reactivity, stability, and selectivity (Fig. 1b). Therefore, the development of a reliable approach for the incorporation of privileged homogeneous bipyridyl-based Ni catalysts into carbon nitride (CN) is in high demand.

Potassium poly(heptazine imide) (K-PHI) exhibits a well-defined and highly ordered structure of negative PHI-layers with K+ cations as charge compensation[57–60], offering the possibility to exchange K+ in matrix[61–63] with cationic Ni complexes. Initial attempts to incorporate

bpyNi$^{2+}$ into semiconductor photocatalysts such as g-C$_3$N$_4$, mpg-CN, and CCN led to quite low Ni-loading (0.05−0.23 wt%), while the use of PHI as host improved the $L_nNi$-loading to as much as 6.0 wt% (Fig. 1c). Herein we describe our efforts toward the use of cation exchange strategy to construct a series of bipyridyl-Ni-functionalized semiconductor materials as a new type of Ni-photo bifunctional solid catalysts. The resulting $L_nNi$-PHI catalysts bearing up to 6 wt% site-isolated bipyridyl-Ni species possess a series of advantages (Fig. 1d) and exhibit exceptional metallaphotocatalytic activity, chemical stability, and recyclability in diverse C−P, C−S, C−O, and C−N cross-coupling reactions with broad substrate scope and good functional group tolerance (Fig. 1f). Upon the visible-light irradiation, the excited catalyst undergoes inner sphere ligand-to-metal charge transfer (LMCT) process to generate the reactive Ni(I) species, thus significantly increasing the catalytic activity over the dual catalytic system that proceeds via outer sphere single electron transfer (SET) between the photocatalyst and nickel catalyst. The resulting Ni(I) species then undergoes oxidative addition with aryl iodide to afford a Ni(III) intermediate, followed by ligand exchange with different types of nucleophiles. Facile reductive elimination at the Ni(III) intermediate delivers

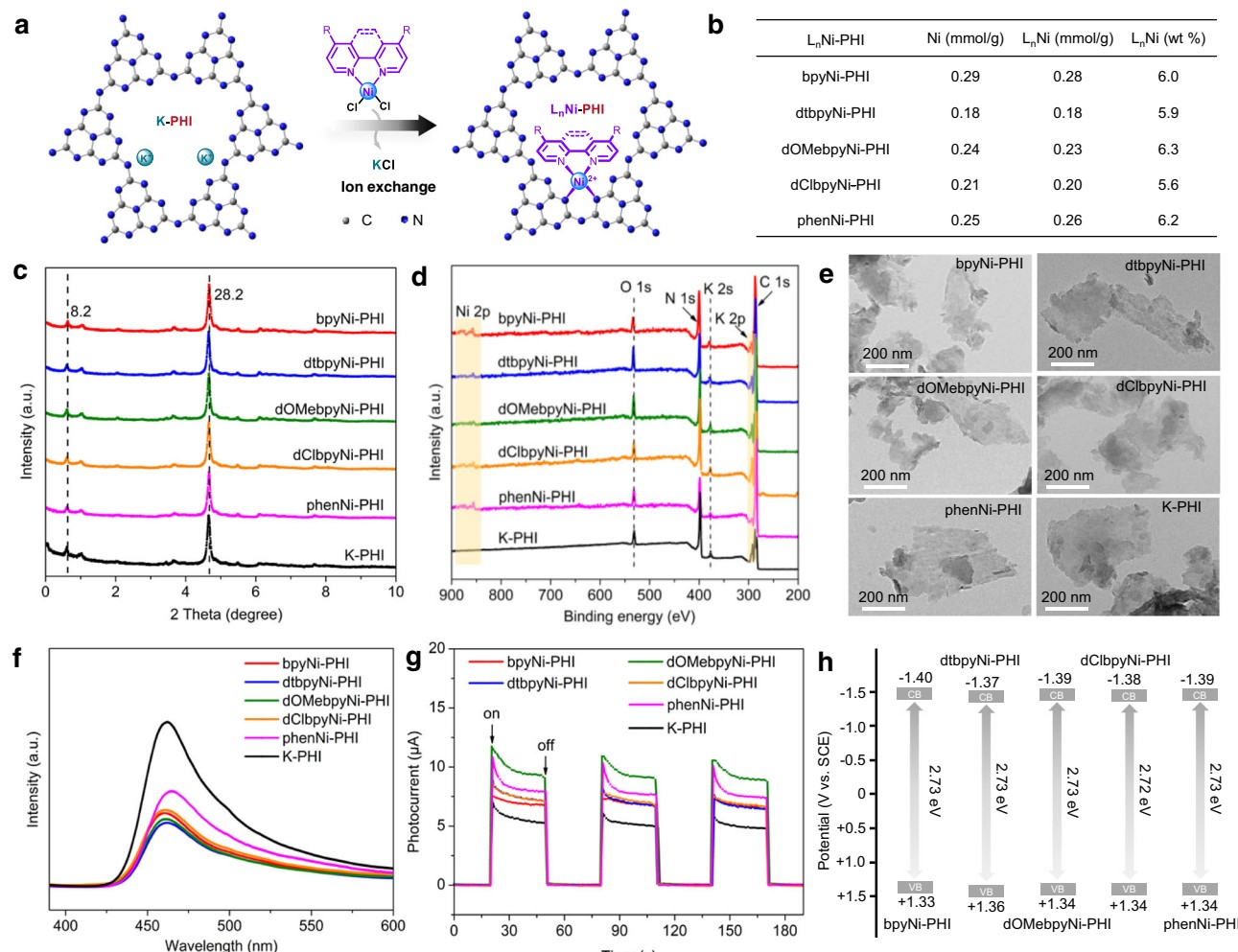

**Fig. 2 | Synthesis and characterizations of $L_nNi$-PHI. a** Schematic illustration for the synthesis of $L_nNi$-PHI catalysts by cation exchange method. **b** Amount of nickel and ligand in $L_nNi$-PHI determined by ICP-OES and $^1$H-NMR analysis. **c** XRD patterns. **d** XPS survey spectra. **e** TEM images. **f** PL spectra. **g** Transient photocurrent response. **h** Schematic drawing of band structures.

the C-heteroatom bond cross-coupling products along with the regeneration of the active Ni(I) species. To be noted, to prevent the thermodynamically favored comproportionation between Ni(I) and Ni(III) species, continuous blue LED irradiation is required to maintain high concentration of Ni(I) species. (Fig. 1e).

## Results

### Preparation of $L_nNi$-PHI catalysts

Following our previously reported modified method, K-PHI, a type of carbon nitride (CN), was prepared by the direct thermal polymerization of melamine in the presence of KCl and NH$_4$Cl salt at 550 °C[64]. A series of nickel(II) complexes of the type $L_nNiCl_2(H_2O)_n$, where $L_n$ (ligand) = bpy (2,2'-bipyridine), dtbpy (4,4'-di-tert-butyl-2,2'-bipyridine), dOMebpy (4,4'-dimethoxy-2,2'-bipyridine), dClbpy (4,4'-dichloro-2,2'-bipyridine) and phen (1,10-phenanthroline), were synthesized by reacting NiCl$_2$·6(H$_2$O) with a 5% excess of ligand in ethanol[65]. The $L_nNi$-PHI catalysts including bpyNi-PHI, dtbpyNi-PHI, dOMebpyNi-PHI, dClbpyNi-PHI, and phenNi-PHI were prepared by cation exchange of K$^+$ in K-PHI (40 mg/mL) by $L_nNi^{2+}$ in $L_nNiCl_2(H_2O)_n$ (8 mg/mL) in DMF under N$_2$ at room temperature for 3 days (Fig. 2a, see experimental details in the Supplementary Information). The excess nickel complexes and other impurities in $L_nNi$-PHI were completely removed through extensive washing with DMF, deionized water, and acetonitrile. According to the

inductively coupled plasma optical emission spectrometry (ICP-OES) results, the content of Ni in $L_nNi$-PHI was determined to be 0.29 mmol/g for bpyNi-PHI, 0.18 mmol/g for dtbpyNi-PHI, 0.24 mmol/g for dOMebpyNi-PHI, 0.21 mmol/g for dClbpyNi-PHI, 0.25 mmol/g for phenNi-PHI, respectively (Fig. 2b). Preliminary leaching experiments of $L_nNi$-PHI in various solvents including DMF, MeCN, DMSO, MeOH, CH$_2$Cl$_2$, THF, and water revealed that the amount of the leached Ni is less than 5% of the total Ni in bpyNi-PHI (Supplementary Table 1). However, the aqueous solution of hydrochloric acid can completely remove Ni$^{2+}$ and K$^+$ of $L_nNi$-PHI (Supplementary Table 2) via cation exchange with H$^+$ to afford crystalline carbon nitride (CCN)[66] bearing neutral PHI 2D-layers. The nickel complexes in the resulted solution after acid treatment of $L_nNi$-PHI can be detected via $^1$H nuclear magnetic resonance ($^1$H-NMR, Supplementary Fig. 1), confirming the presence of Ni complexes in $L_nNi$-PHI. Based on further $^1$H-NMR analyses and ICP-OES results, the ratio between Ni and ligand and the loading of $L_nNi$ in $L_nNi$-PHI were unambiguously determined to be ~1/1 and ~6 wt %, respectively (Fig. 2b and Supplementary Table 3). Additionally, it was found that K$^+$ in K-PHI was released and replaced by $L_nNi^{2+}$ at a molar ratio close to 2:1 during the exchange (Supplementary Fig. 2 and Supplementary Table 4), while the majority of K ions (~80%) remain in the resulted $L_nNi$-PHI as charge compensation (Supplementary Tables 3 and 4).

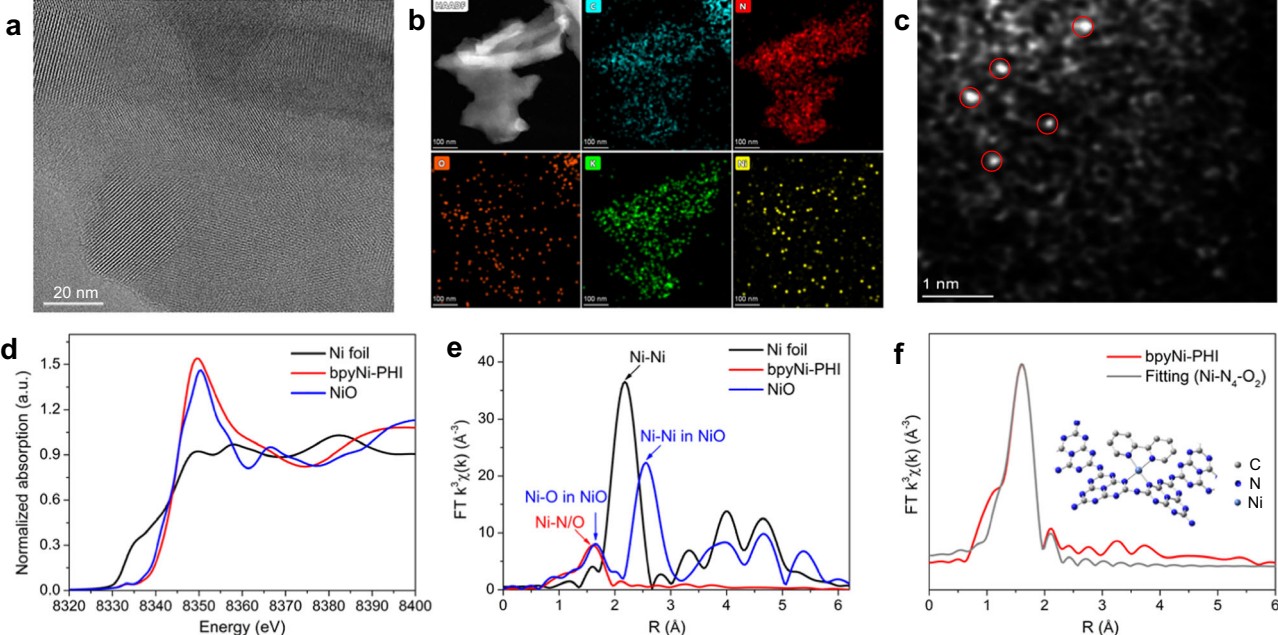

**Fig. 3 | Structural characterizations of bpyNi-PHI. a** HRTEM image of bpyNi-PHI. **b** Elemental mapping images of bpyNi-PHI. **c** HAADF-STEM image of bpyNi-PHI. **d** Ni K-edge XANES spectra. **e** FT-EXAFS spectra of Ni foil, bpyNi-PHI, and Ni-O. **f** The corresponding EXAFS fitting curves of bpyNi-PHI (inset: simulated structure model).

## Structure Characterization

Fourier-transform infrared (FT-IR) spectra of L$_n$Ni-PHI (Supplementary Fig. 3) show typical absorption bands of PHI at 1200–1800 cm$^{-1}$, 810 cm$^{-1}$, 998 cm$^{-1}$, and 917 cm$^{-1}$, which are ascribed to the stretching vibration of CN heterocycles, bending vibration of the heptazine rings, and C–N vibration signal, respectively[57–60], suggesting the introduction of L$_n$Ni did not change the chemical structure of PHI. UV-vis diffuse reflectance spectra (DRS) of L$_n$Ni-PHI reveal the absorption region of the materials up to 480 nm with band gaps -2.72–2.73 eV, which are comparable to that of K-PHI (Supplementary Fig. 4 and 5). The X-ray powder diffraction (XRD) patterns of L$_n$Ni-PHI present two typical diffraction peaks at 8.2° (110) and 28.2° (002), reflecting the in-plane ordering of heptazine motifs and interlayer stacking of aromatic systems in the PHI structure[57] are well maintained during L$_n$Ni incorporation (Fig. 2c). The X-ray photoelectron spectra (XPS) indicate that K-PHI comprise elements of C, N, O, and K, while L$_n$Ni-PHI consists of C, N, O, K, and Ni (Fig. 2d and Supplementary Figs. 6–11). The binding energy located at 855.9 (Ni 2p$_{3/2}$) and 873.5 eV (Ni 2p$_{1/2}$) is assigned to Ni$^{2+}$[53,54]. Transmission electron microscopy (TEM) images suggest that L$_n$Ni-PHI is a layered structure with nanometer-sized domains, which is similar to that of K-PHI (Fig. 2e). The photoluminescence (PL) spectra of L$_n$Ni-PHI display reduced emission (Fig. 2f), indicative of efficient electron transfer or energy transfer from the emissive state to Ni(II), producing the Ni(I) or excited Ni(II) species[67,68]. The shorter PL lifetime ($\tau = 0.34–0.42$ ns for L$_n$Ni-PHI vs $\tau = 0.57$ ns for K-PHI, Supplementary Fig. 12) and stronger photocurrent density (Fig. 2g) further indicates good charge separation and migration in L$_n$Ni-PHI, which is beneficial to potential photocatalytic applications. The conduction bands (CB) of L$_n$Ni-PHI are determined to be approx. −1.40 V (vs SCE) from Mott-Schottky plots (Fig. 2h and Supplementary Fig. 14), which is more negative than the reduction potential of Ni$^{II}$/Ni$^I$ (−0.93 V vs SCE)[33] or Ni$^{II}$/Ni$^0$ (−1.36 V vs SCE)[69], suggesting the feasibility of L$_n$Ni-PHI in photo/Ni dual catalysis.

To gain more structural insight and elucidate the electronic and microstructural information of Ni atoms in L$_n$Ni-PHI, we took bpyNi-PHI as a representative to conduct high-resolution transmission electron microscopy (HRTEM), high-angle annular dark-field scanning transmission electron microscopy (HAADF-STEM) and X-ray absorption spectroscopy (XAS) analysis. HRTEM image suggests bpyNi-PHI has good crystallinity with intrinsic crystal facet and obvious lattice fringes (Fig. 3a). Further elemental mapping reveals the uniform distribution of C, N, O, K, and Ni elements in bpyNi-PHI (Fig. 3b). The HAADF-STEM image of bpyNi-PHI (Fig. 3c) confirms the presence of single-atoms without observation of metal particles or clusters. From the Ni K-edge X-ray absorption near-edge structure (XANES) spectra (Fig. 3d), the absorption edge position and spectral line shape of bpyNi-PHI closely resemble those of Ni-O, indicating the oxidation state of the Ni single-atoms close to +2. According to the linear combination fittings on XANES profiles, the average chemical valence of Ni is calculated to be +2.03 (Supplementary Fig 15 and Supplementary Table 6), which is in good agreement with the Ni 2p XPS results. The Fourier transformation (FT) of Ni K-edge extended X-ray absorption fine structure (EXAFS) spectra of bpyNi-PHI exhibits a prominent peak centered at 1.6 Å for Ni-N/O coordination, while no Ni-Ni and Ni-O-Ni characteristic peaks are observed at 2.2 Å (Ni foil) and 2.9 Å (Ni-O-Ni structure), suggesting the atomically dispersion of Ni atoms (Fig. 3e)[54,70]. The EXAFS fitting results indicate that the Ni center adopts Ni-N$_4$-O$_2$ structure with the coordination number of -6 (Fig. 3f and Supplementary Table 6). As the EXAFS cannot differentiate the contribution from coordinated N and O atoms due to their similar scattering factors, the existence of Ni-O bonds attributed to water cannot be excluded. Taken together, we propose Ni in bpyNi-PHI might be bonded with adjacent pyridinic nitrogen of two separated triazine units and one bipyridine molecule (inset of Fig. 3f), which is also in agreement with the structure preliminarily optimized by DFT calculation (Supplementary Fig. 16).

## Catalytic activity of L$_n$Ni-PHI

In combination with the above analyses, we speculate that the obtained L$_n$Ni-PHI materials feature site-isolated active L$_n$Ni$^{2+}$ species and meanwhile retain the photocatalytic activity of the parent PHI, thus showing great potential to be served as highly effective heterogeneous metallaphotocatalysts for promoting visible-light-mediated organic transformations. In order to verify our hypothesis, five

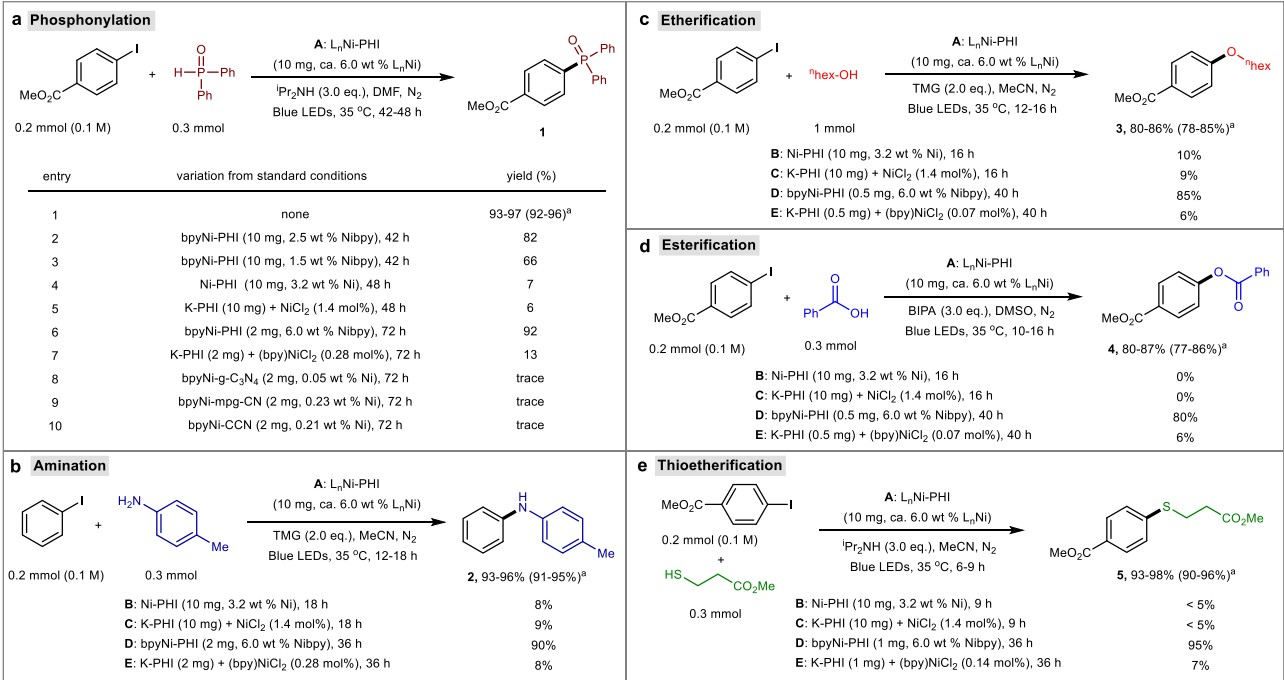

**Fig. 4 | Study on catalytic activity of $L_nNi$·PHI.** All reactions were conducted at 0.2 mmol scale under $N_2$ atmosphere and blue LEDs irradiation (24 W, 460 ± 5 nm) without extra heating (at 35 ± 5 °C). Yields of products **1–5** were determined by

[1]H-NMR analysis using 1,3,5-trimethoxybenzene as an internal standard. **a** Phosphonylation. [a]Isolated yield in parenthesis. **b** Amination. **c** Etherification. **d** Esterification. **e** Thioetherification.

representative and synthetic useful carbon-heteroatom bond formation reactions including phosphonylation, amination, etherification, esterficiation, and thioetherification of aryl iodide with corresponding heteroatom coupling partners were investigated (Fig. 4a–e). After considerable efforts (see optimization in Supplementary Tables 7–11 of Supplementary Information), we found that $L_nNi$·PHI (~6 wt % $L_nNi$) including bpyNi·PHI, dtbpyNi·PHI, dOMebpyNi·PHI, dClbpyNi·PHI, and phenNi·PHI could effectively catalyze all five cross-coupling reactions, providing high yields of the desired C–P, C–N, C–O and C–S coupling products **1–5** in the presence of suitable base and solvent under blue light irradiation (Fig. 4a–e, condition A). Compared to other $L_nNi$·PHI catalysts, dClbpyNi·PHI exhibits a relatively lower catalytic activity, requiring slightly longer reaction time (Supplementary Tables 7–11). These reactions failed to proceed in the absence of $L_nNi$·PHI catalysts, light, or base additive, implying all were crucial for these transformations (Supplementary Tables 7–11). We further conducted a series of control experiments to probe the function of different components in $L_nNi$·PHI and their synergy mechanism.

The bpyNi·PHI-based catalysts bearing lower bpyNi loadings (2.5 wt % and 1.5 wt %) delivered product **1** in decreased yields (Fig. 4a, entries 2 and 3), signifying that increasing the number of $LnNi^{2+}$ active sites in $L_nNi$·PHI is beneficial for the activity. Only trace amounts or low yields of coupled products were observed with Ni·PHI (3.2 wt % Ni, prepared from $NiCl_2$ and K·PHI without additional bpy ligand) or a mix of both K·PHI and $NiCl_2$ (Fig. 4a, entries 4 and 5; Fig. 4b–e, conditions B and C), verifying the bidentate nitrogen ligands (L) in $L_nNi$·PHI play an essential role in dictating the excellent activity. Remarkably, with very low loadings of bpyNi·PHI (0.28 mol%, 0.28 mol%, 0.07 mol%, 0.07 mol%, and 0.14 mol% based on bpyNi for phosphonylation, amination, etherification, esterification and thioeserification, respectively), the reactions proceeded smoothly to afford C–P, C–N, C–O and C–S coupling products **1–5** with turnover numbers (TONs) of ~330, ~320, ~1200, ~1100 and ~680, respectively (Fig. 4a, entry 6; Fig. 4b–e, condition D). In contrast, K·PHI with separate addition of bpyNiCl₂ gave low yields of coupling products (**1**: 13%, **2**: 8%, **3**: 6%, **4**: 6%, and **5**:

7%) at the same catalyst loading under identical conditions (Fig. 4a, entry 7; Fig. 4b–e, condition E), demonstrating that the proximity and cooperativity of the $LnNi^{2+}$ active species and PHI photocatalyst carrier in $L_nNi$·PHI might facilitate SET, and free radical transfer. In addition, other $L_nNi$·CN catalysts exhibited much lower catalytic activities, affording coupling product **1** in low yields, due to the limited Ni-loading in these catalysts (Fig. 4a, entries 8–10).

## Recyclability and leaching test

Apart from the enhanced catalytic activity, another intrinsic advantage of $L_nNi$·PHI is potential reusability of the solid metallaphotocatalyst. Therefore, we conducted the recyclability and leaching test to probe the multinuclear catalyst deactivation and the heterogeneity of the reaction. As shown in Fig. 5a, the recovered bpyNi·PHI can be reused for further cycles to give C–P coupling product **1** and the rates of reactions over five catalytic cycles remain the same. Meanwhile, bpyNi·PHI can also at least be recycled five times without loss of activity in C–N, C–O, and C–S couplings, affording the corresponding product **2-5** with maintained yields (Fig. 5b). The recovered bpyNi·PHI after photocatalytic C–P coupling reaction was characterized by UV-vis DRS, IR, PXRD, XPS, TEM, and NMR to demonstrate the robustness of the catalyst. After catalysis, two typical diffraction peaks at 8.2° and 28.2° in XRD patterns (Fig. 5c) and the Ni 2p peak at 856 eV in XPS survey spectra (Fig. 5d and Supplementary Fig. 21) remain unchanged, indicating the structure of bpyNi·PHI was preserved during the catalysis. The FT-IR and UV-vis DRS spectra of bpyNi·PHI were also well maintained before and after the reaction (Supplementary Fig. 22). According to the statistical results of TEM images (Fig. 5e and Supplementary Fig. 23), bpyNi·PHI maintained the layered structure without formation of agglomerated Ni/Ni-O nanoparticles in the catalytic process[56]. Furthermore, the nickel complex in the recovered bpyNi·PHI with Ni to bpy molar ratio of ~1/1 was detected by [1]H-NMR in combination with the ICP-OES results (Fig. 5f, g). Besides, slight leakage of $bpyNi^{2+}$ catalytic species was observed during the recycling (Fig. 5g).

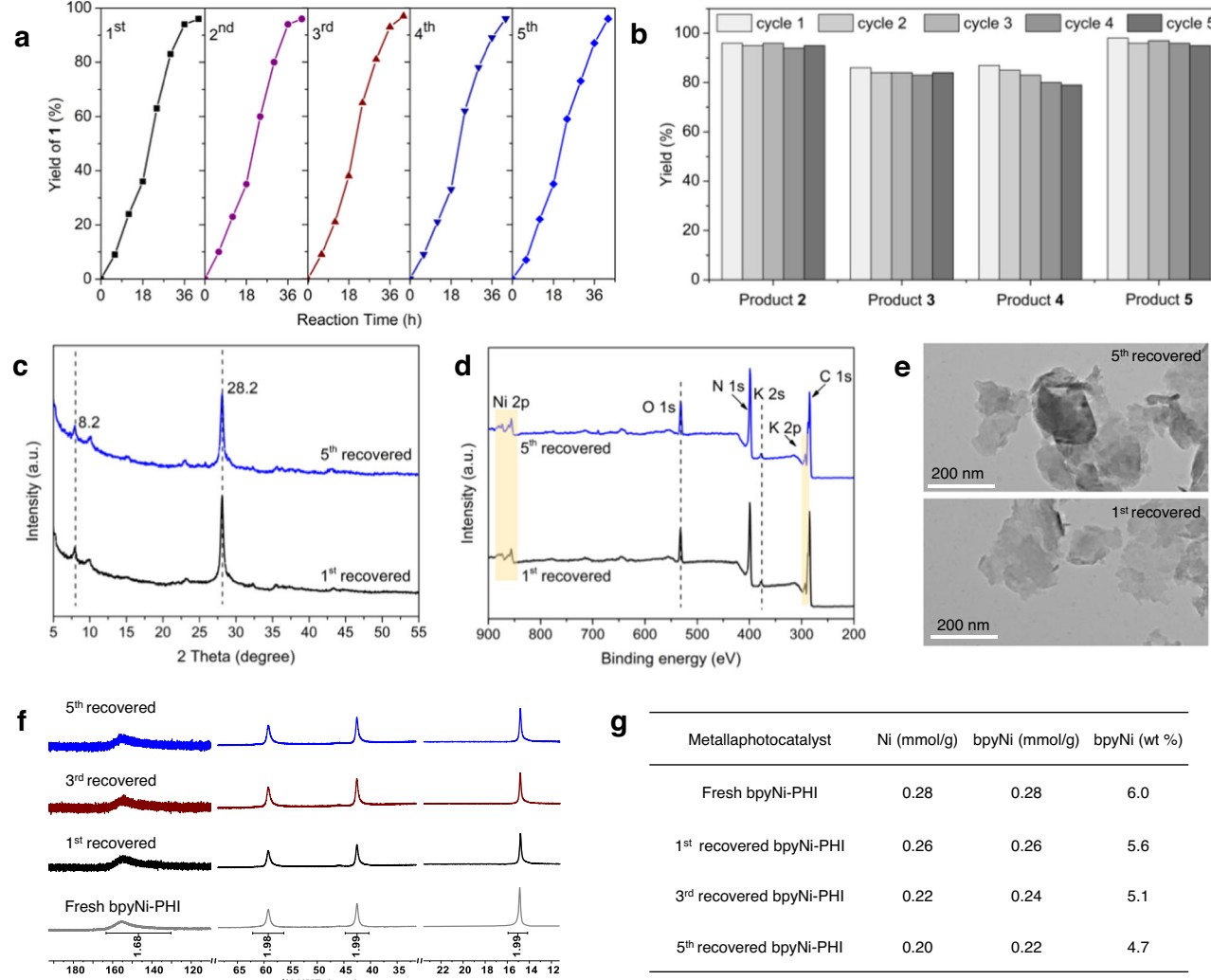

**Fig. 5 | Recyclability and leaching test. a** Kinetic profile of the photocatalytic C−P coupling (0.2 mmol scale with 10 mg of bpyNi-PHI) over five catalytic cycles. **b** Catalyst recycling (for five catalytic cycles) of the C−N, C−O, and C−S coupling. **c** XRD patterns. **d** XPS survey spectra. **e** TEM images of recovered bpyNi-PHI catalyst after photocatalytic C−P coupling reaction. **f** $^1$H-NMR spectra (DMSO-$d6$, 400 MHz) of recovered bpyNi-PHI after treatment with 1.5 M HCl in a mixed H$_2$O:MeOH (1:1, v-v) solvent. **g** Amount of nickel and ligand in recovered bpyNi-PHI determined by ICP-OES and $^1$H-NMR analysis.

## Substrate scope of bpyNi-PHI catalyzed C−heteroatom couplings

Encouraged by the excellent catalytic activity, reliable stability, and good recyclability of L$_n$Ni-PHI, we sought to explore the scope and robustness of the bpyNi-PHI based heterogeneous metallaphotocatalyst in catalyzing diverse C−heteroatom cross-couplings. As shown in Fig. 6, a wide range of aryl/heteroaryl iodides bearing various electron-donating and electron-withdrawing substituents could undergo C−P, C−N, and C−S couplings smoothly, affording the corresponding triarylphosphine oxides (**2**, **6**−**20**), biaryl/heteroaryl amines (**22**−**34**), *N*-aryl/heteroaryl sulfonamides (**35**−**39**), thioethers (**86**−**93**) for good to excellent yields (70−98%).

A series of synthetically useful functional groups, including ketone (**6**, **29**, **61**), aldehyde (**92**), ester (**1**, **3**, **5**, **28**), cyano (**7**, **27**, **62**), hydroxyl (**13**, **90**), and even unprotected NH$_2$ groups (**14**, **91**), are compatible with the reaction conditions. Regarding C−O couplings, this catalytic system works well with electron-deficient aryl iodides and heteroaryl iodide (**3**, **4**, **61**−**63**, **66**), whereas substrates lacking an electron-withdrawing group exhibit low reactivity (**64**, **65**), presumably due to the issue of oxidative addition of the corresponding aryl iodides with Ni species. Notably, the couplings could proceed selectively at the iodo-

functionalized carbon atom; both aryl bromide and chloride bonds remained intact, thus providing the possibility for further synthetic elaborations (**8**, **9**, **25**, **30**, **37**, **86**). Additionally, strong electron-deficient aryl bromide is also suitable for C−P, C−O, and C−N couplings with lower reactivity (**1**, **3**, **28**).

With respect to the heteroatom coupling partners, a wide array of other *P-, N-, O-, S*-containing weak nucleophiles can be effectively coupled, including dialkyl phosphite (**21**), aryl/heteroaryl amines (**2**, **40**−**48**), primary alkyl amines (**49**−**51**), sulfonamides (**54**, **55**), sulfoximine (**56**), imine (**57**), amide (**58**), carbamates (**59**, **60**), primary alcohols (**67**−**75**), secondary alcohols (**76**, **77**), water (**78**), aromatic acids (**79**, **80**), aliphatic acids (**81**, **82**), olefinic acids (**83**−**85**), aliphatic thiol (**94**), aromatic thiol (**95**) and thiolacids (**96**). However, secondary amines participated in this reaction with poor efficiency and selectivity, affording the desired C−N coupled products (**52** and **53**) in low yields, along with the formation of dehalogenated and phenol by-products. As in the case of aryl iodides, the coupling reactions exhibited good compatibility with functional groups such as cyclopropyl, benzyl, alkenyl, alkynyl groups (**69**, **70**, **73**, **74**). In addition, high chemoselectivities for C−N and C−S coupling were observed in the presence of alcoholic functionality (**46**, **94**).

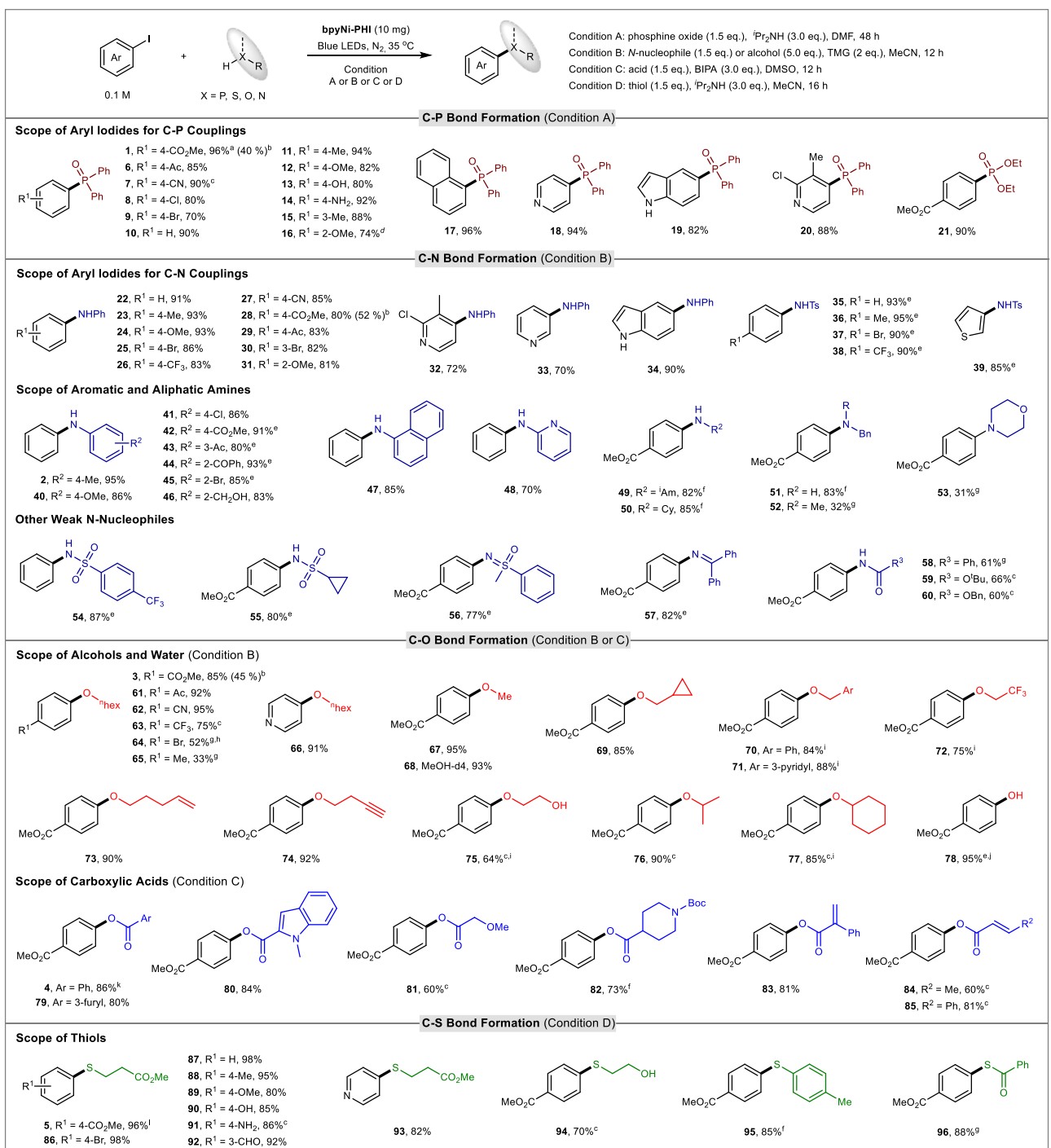

**Fig. 6 | Substrate scope of bpyNi·PHI catalyzed C–P, C–N, C–O, and C–S couplings.** Reaction conditions: aryl iodide (0.2 mmol), bpyNi·PHI (10 mg, 6 wt %), phosphine oxide (0.3 mmol, 1.5 eq.) or alcohol (1.0 mmol, 5.0 eq.) or amine (0.3 mmol, 1.5 eq.) or thiol (0.3 mmol, 1.5 eq). Condition A: $^{i}$Pr$_2$NH (0.6 mmol, 3.0 eq.) in DMF (2 mL). Condition B: TMG (0.4 mmol, 2.0 eq.) in MeCN (2 mL). Condition C: BIPA (0.6 mmol, 3.0 eq.) in DMSO (2 mL). Condition D: $^{i}$Pr$_2$NH

(0.6 mmol, 3.0 eq.) in MeCN (2 mL). [a]42 h. [b]Using methyl 4-bromobenzoate as coupling partner. [c]24 h. [d]60 h. [e]16 h. [f]36 h. [g]48 h. [h]13/1 ratio of bromo-substituted product **64** vs iodo-substituted product. [i]Alcohol (3.0 eq.). [j]Using 10 eq. of H$_2$O as coupling partner. [k]10 h. [l]6 h. All reactions were performed under N$_2$ atmosphere and blue LEDs irradiation (24 W, 460 ± 5 nm) without extra heating (at 35 ± 5 °C). Isolated yields are given.

## Late-stage diversification and gram-scale synthesis

The applicability of bpyNi·PHI-based heterogeneous metallaphotocatalytic C–heteroatom bond formation was further explored for the late-stage diversification of bioactive and pharmaceutical molecules. As shown in Fig. 7a, complex aryl iodides derived from natural molecules including *L*-menthol and *L*-phenylalanine could undergo diverse C–heteroatom couplings smoothly, delivering the corresponding

ether (**97**), ester (**98, 99**), phenol (**100**), amine (**101**), sulfonamide (**102**), phosphine oxide (**103, 105**) and thioether (**104**) in satisfactory yields. A series of natural α-amino acid derivatives such as *L*-alanine, *L*-phenylalanine, *L*-proline, *L*-serine, *L*-cysteine were amenable to the C–O and C–S couplings to form the corresponding esterification, etherification, and thioletherification products (**106–108, 109, 110**) with high efficiency. The carbohydrate alcohols derived from

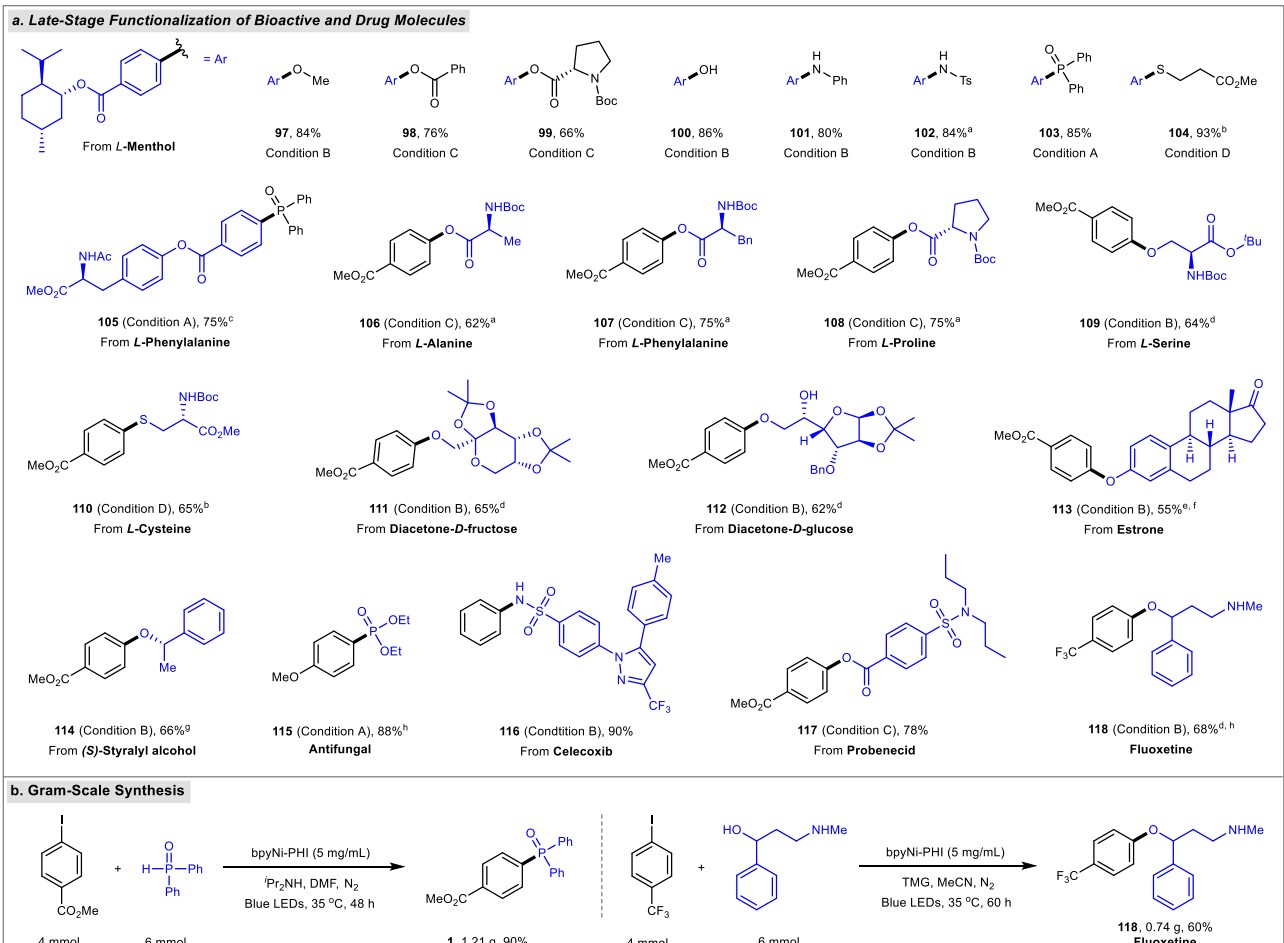

**Fig. 7 | Late-stage diversification of bioactive compounds and pharmaceutical molecules. a** Late-stage functionalization of bioactive and drug molecules. [a]16 h. [b]12 h. [c]36 h. [d]Alcohol (0.3 mmol, 1.5 eq.). [e]Using elaborate phenol (estrone) as coupling partner. [f]72 h. [g]Alcohol (0.6 mmol, 3.0 eq.). [h]60 h. Isolated yields are given. **b** Gram-scale synthesis. See Fig. 6 and Supplementary Information for detailed reaction conditions and procedures.

*D*-fructose and *D*-glucose worked well under current catalytic system to produce the desired *O*-arylated products (**111**, **112**) in good yields. Elaborated phenol (estrone) and (*S*)-styrylyl alcohol also proved to be suitable *O*-containing coupling partners (**113**, **114**). To further illustrate the potential practicality, we successfully applied the developed protocol to the preparation of antifungal phosphonate (**115**), derivatives of anti-inflammatory drug celecoxib (**116**), and antigout drug probenecid (**117**) as well as antidepressant drug fluoxetine (**118**). Moreover, the scalability of our protocol was demonstrated by gram-scale synthesis of triarylphosphine oxide **1** and antidepressant drug fluoxetine **118**. As shown in Fig. 7b, the reaction scale was increased 20-fold in batches to produce the coupling product with only a minimal decrease in yield.

In summary, we have developed a facile cation exchange strategy to incorporate a series of privileged bipyridyl-based Ni catalysts into highly ordered and crystalline K-PHI. A variety of PHI-supported cationic bipyridyl-based Ni catalysts have been successfully constructed and fully characterized by NMR, ICP-OES, XPS, HAADF-STEM and XAS. The obtained $L_nNi$-PHI solid catalysts, featuring high dosage (~6 wt %) of site-isolated bipyridyl-Ni active species with Ni to bpy molar ratio of ~1/1, can be served as highly effective and recyclable metallaphotocatalysts for diverse C–heteroatom cross-coupling reactions with broad substrate scope and good functional group tolerance. The practicability of these heterogeneous protocols has also been demonstrated in the late-stage diversification of various bioactive compounds and pharmaceutical molecules. Notably, the proximity between Ni and photocatalytic centers in $L_nNi$-PHI significantly

enhances the photo/Ni dual catalytic activity, thus dramatic increasing the TONs (300–1200 for Ni). Additionally, the heterogeneous $L_nNi$-PHI catalysts exhibit exceptional chemical stability with low Ni leaching during the reaction and thus can be recycled. We anticipate the ligand exchange strategy report here will provide the basis for developing other novel PHI-supported metallaphotocatalysts.

## Methods
### Preparation of $L_nNi$-PHI
To a 20 mL oven-dried sealed tube equipped with a magnetic stir bar was added $(bpy)NiCl_2(H_2O)_n$ (40 mg) and K-PHI (200 mg). It was capped with a rubber septum, evacuated, and backfilled with nitrogen three times. Then, DMF (5 mL) was added via syringe. The mixture was stirred under nitrogen at room temperature for 72 h and centrifuged. The resulting solid was successively washed with DMF (2 × 5 mL) with sonication and separation by centrifugation, deionized water (2 × 5 mL) with sonication and separation by centrifugation, and MeCN (2 × 5 mL) with sonication and separation by centrifugation. The resulting powder was dried at 50 °C under vacuum for 12 h to yield bpyNi-PHI as dark yellow powder. (average yield per batch: ~202 mg). Other $L_nNi$-PHI catalysts including dtbpyNi-PHI, dOMebpyNi-PHI, dClbpyNi-PHI and phenNi-PHI were prepared using the corresponding bipyridyl-Ni(II) complexes instead of $(bpy)NiCl_2(H_2O)_n$ via the same procedure as bpyNi-PHI. According to the ICP-OES results, the content of Ni in $L_nNi$-PHI was determined to be 0.29 mmol/g (1.68 wt %) for bpyNi-PHI, 0.18 mmol/g (1.04 wt %) for dtbpyNi-PHI, 0.24 mmol/g

(1.39 wt %) for dOMebpyNi-PHI, 0.21 mmol/g (1.22 wt %) for dClbpyNi-PHI, 0.25 mmol/g (1.45 wt %) for phenNi-PHI, respectively. The scale-up synthesis was also performed using 2.0 g of K-PHI and 0.4 g of (bpy)NiCl$_2$(H$_2$O)$_n$ in DMF (50 mL) to produce 2.1 g of bpyNi-PHI with 1.53 wt % Ni content.

### General procedure 1 for bpyNi-PHI based heterogeneous photocatalytic C−P couplings
To a 10 mL oven-dried sealed tube equipped with a magnetic stir bar was added the corresponding aryl iodide (0.2 mmol, 1.0 eq.), H-phosphine oxide (0.3 mmol, 1.5 eq.) and bpyNi-PHI (10 mg, 6.0 wt % Nibpy). Then, dry DMF (2 mL) and $^i$Pr$_2$NH (0.6 mmol, 3.0 eq.) were added. The tube was closed with a rubber septum and the reaction mixture was degassed by three cycles vacuum/N$_2$ of "freeze-pump-thaw". The reaction mixture was stirred and irradiated by blue LEDs (24 W, 460 ± 5 nm) without extra heating (35 ± 5 °C) for the indicated time. In each case, the blue LEDs was placed 3 cm from the reaction tube (Supplementary Fig. 19a). An independent fan was used to maintain the temperature inside the irradiation reaction system. Upon completion, the reaction mixture was diluted with deionized water (5 mL) and extracted with ethyl acetate (3 × 5 mL). The combined organic layer was washed with brine, dried over anhydrous Na$_2$SO$_4$, and concentrated. Finally, the crude residue was purified by silica gel column chromatography. For comparison, two 40 W Kessil PR lamp (50% power, 456 nm) were used as alternative light sources (Supplementary Fig. 19b), similar yield of methyl 4-(diphenylphosphoryl)benzoate (**1**, 48 h, 63.2 mg, 94%,) as obtained.

### General procedure 2 for bpyNi-PHI based heterogeneous photocatalytic C−N couplings
To a 10 mL oven-dried sealed tube equipped with a magnetic stir bar was added the corresponding aryl iodide (0.2 mmol, 1.0 eq.), amine (0.3 mmol, 1.5 eq.), and bpyNi-PHI (10 mg, 6.0 wt % Nibpy). Then, dry MeCN (2 mL) and TMG (0.4 mmol, 2.0 eq.) were added. The tube was closed with a rubber septum and the reaction mixture was degassed by three cycles vacuum/N$_2$ of "freeze-pump-thaw". After that the reaction mixture was stirred and irradiated by blue LEDs (24 W, 460 ± 5 nm) without extra heating (35 ± 5 °C) for the indicated time. An independent fan was used to maintain the temperature inside the irradiation reaction system. In each case, the blue LEDs was placed 3 cm from the reaction tube (Supplementary Fig. 19a). Upon completion, the reaction mixture was concentrated under reduced pressure to evaporate the solvent, and the crude residue was purified by silica gel column chromatography.

### General procedure 3 for bpyNi-PHI based heterogeneous photocatalytic C−O couplings
To a 10 mL oven-dried sealed tube equipped with a magnetic stir bar was added the corresponding aryl iodide (0.2 mmol, 1.0 eq.), alcohol (1.0 mmol, 5.0 eq.), and bpyNi-PHI (10 mg, 6.0 wt % Nibpy). Then, dry MeCN (2 mL) and TMG (0.4 mmol, 2.0 eq.) were added. The tube was closed with a rubber septum and the reaction mixture was degassed by three cycles vacuum/N$_2$ of "freeze-pump-thaw". The reaction mixture was stirred and irradiated by blue LEDs (24 W, 460 ± 5 nm) without extra heating (35 ± 5 °C) for the indicated time. In each case, the blue LEDs was placed 3 cm from the reaction tube (Supplementary Fig. 19a). An independent fan was used to maintain the temperature inside the irradiation reaction system. Upon completion, the reaction mixture was concentrated under reduced pressure to evaporate the solvent, and the crude residue was purified by silica gel column chromatography.

### General procedure 4 for bpyNi-PHI-based heterogeneous photocatalytic C−O couplings
To a 10 mL oven-dried sealed tube equipped with a magnetic stir bar was added the corresponding aryl iodide (0.2 mmol, 1.0 eq., if solid), carboxylic acid (0.3 mmol, 1.5 eq., if solid), and bpyNi-PHI (10 mg, 6.0 wt % Nibpy). Then, dry DMSO (2 mL) and BIPA (0.6 mmol, 3.0 eq.) were added. The tube was closed with a rubber septum and the reaction mixture was degassed by three cycles vacuum/N$_2$ of "freeze-pump-thaw". After that the reaction mixture was stirred and irradiated by blue LEDs (24 W, 460 ± 5 nm) without extra heating (35 ± 5 °C) for the indicated time. In each case, the blue LEDs was placed 3 cm from the reaction tube (Supplementary Fig. 19a). An independent fan was used to maintain the temperature inside the irradiation reaction system. Upon completion, the reaction mixture was diluted with deionized water (5 mL) and extracted with ethyl acetate (3 × 5 mL). The combined organic layer was washed with brine, dried over anhydrous Na$_2$SO$_4$, and concentrated. Finally, the crude residue was purified by silica gel column chromatography.

### General procedure 5 for bpyNi-PHI-based heterogeneous photocatalytic C−S couplings
To a 10 mL oven-dried sealed tube equipped with a magnetic stir bar was added the corresponding aryl iodide (0.2 mmol, 1.0 eq.), thiol or thiolacid (0.3 mmol, 1.5 eq.), and bpyNi-PHI (10 mg, 6.0 wt % Nibpy). Then, dry MeCN (2 mL) and $^i$Pr$_2$NH (0.6 mmol, 3.0 eq.) were added. The tube was closed with a rubber septum and the reaction mixture was degassed by three cycles vacuum/N$_2$ of "freeze-pump-thaw". After that the reaction mixture was stirred and irradiated by blue LEDs (24 W, 460 ± 5 nm) without extra heating (35 ± 5 °C) for the indicated time. In each case, the blue LEDs was placed 3 cm from the reaction tube (Supplementary Fig. 19a). An independent fan was used to maintain the temperature inside the irradiation reaction system. Upon completion, the reaction mixture was concentrated under reduced pressure to evaporate the solvent, and the crude residue was purified by silica gel column chromatography.

## Data availability
The authors declare that all data generated in this study are available within the article and the Supplementary Information. Any additional detail can be requested from the corresponding authors.

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

## Acknowledgements

This work is supported by funding from the Natural Science Foundation of Chongqing (CSTB2022NSCQ-MSX1105 to Y.C., CSTB2022NSCQ-MSX1032 to Y.T.), the National Natural Science Foundation of China (Grant no. 21801030 to Y.C.), the King Abdullah University of Science and Technology (KAUST), Saudi Arabia, Office of Sponsored Research (URF/1/4025 to C.Z. and M.R.). We thank the Analytical and Testing Center of Chongqing University for assistance with NMR spectrum analysis.

## Author contributions

L.X and Q.Y. performed photocatalytic experiments; Q.Y. and Y.B. synthesized and characterized LnNi-PHI catalysts; C.Z. checked data and carried out DFT calculation; Y.T., M.R., and Y.C. conceived and supervised the research study. Y.C., C.Z., and M.R. wrote the paper with input from all authors. All authors discussed the results. L.X., Q.Y., and C.Z. contributed equally.

## Competing interests

The authors declare no competing interests.
