## [Peer Review File · Nature Communications]

Poly(Heptazine Imide) Ligand Exchange Enables Remarkable Low Catalyst Loadings in Heterogeneous MetallaphotocatalysisREVIEWER COMMENTS

Reviewer #1 (Remarks to the Author):

This study developed a series of heterogeneous bipyridyl-Ni-functionalized PHI (LnNi-PHI) with high Ni loading dosage via a facile cation exchange strategy. The structure of the bpyNi-PHI site constructed on PHI support was thoroughly examined, and its mononuclear nature/configuration was clearly established. Significantly, LnNi-PHI offered activities/TON superior to analogue catalysts (e.g., Ni-PHI, LnNi-mpgCN or semiheterogeneous LnNi/PHI) for diverse cross-coupling reactions (C-N, C-P, C-O and C-S). The catalysts were also recyclable and robust, which are desirable for practical implementation. LnNi-PHI demonstrated impressive substrate scopes that is compatible with electron deficient aryl iodide or low nucleophilic coupling partners. In addition, late-stage diversification of bioactive molecules and pharmaceuticals was successfully accomplished. The design concept/approach of LnNi on PHI is novel, offering exceptional catalytic performance. The reviewer believe this paper would be appeal to broad research communities.

Specific comments.

1. The surface areas and porosities of PHI before and after LnNi exchange would be helpful.
2. EXAFS gave a coordination number of six. However a four-coordinate Ni was shown (from DFT calculation?). Could DFT locate a stable six-coordinate Ni? For example, with solvent or aqua ligand as the author suggested.
3. Could the author provide more discussion on the exceptional activity of LnNi-PHI? For example, a higher loading of Ni on PHI than those on other carbon nitride, or the difference in electronic structure/intrinsic activity of Ni (oxidation states and coordination sphere/bond lengths)
4. What about the activity comparison between LnNi-PHI and integrated Ni-mpgCN (reported by Reisner's group, not Ni-PHI in the manuscript) under the same condition?
5. The conversion/yield obtained using semiheterogenous is PHI/NiLn was rather low; however the reported analogue systems exhibit much better performance. Could the authors comment on this?
6. The intact of C-Br during reaction is somehow surprising. Please comment
7. It is not clear why LnNi-PHI works better in coupling with primary amine than that with secondary, which is contrary to observations made for other Ni/photoredox catalysts.

Reviewer #2 (Remarks to the Author):

This manuscript reports the use of Ni-functionalized Poly(Heptazine Imide) materials as photocatalysts in a variety of C-P, C-S, C-O, and C-N cross-coupling reactions. The materials are carefully characterized, and the catalytic study comprehensively demonstrate the usefulness of the system reported. Therefore, I recommend its publication after consideration of the following minor aspects:

- Figure 1 seems too large.
- 1s N XPS of the materials could be informative specially if compared with the pristine PHI material (Which is not presented).

- Ni-PHI is mentioned in the main text but to understand its nature the reader should go to supplementary information. A very brief explanation in the main text would be appreciated.
- In order to discard homogeneous catalytic processes, an additional experiment is necessary: After one catalytic run, the material is separated by filtration and the formation of products under the reaction conditions is discarded in the absence of material.
- Some comments on the catalytic mechanism would enrich the manuscript.
- The description of the experimental set-up in supplementary information should include not only the wavelength of light source (Blue LEDs) but also its intensity. Also, how constant temperature is granted in the photocatalytic experiment?

Response to Reviewer #1:

Reviewer #1 (Remarks to the Author):

This study developed a series of heterogeneous bipyridyl-Ni-functionalized PHI (LnNi-PHI) with high Ni loading dosage via a facile cation exchange strategy. The structure of the bpyNi-PHI site constructed on PHI support was thoroughly examined, and its mononuclear nature/configuration was clearly established. Significantly, LnNi-PHI offered activities/TON superior to analogue catalysts (e.g., Ni-PHI, LnNi-mpgCN or semiheterogeneous LnNi/PHI) for diverse cross-coupling reactions (C-N, C-P, C-O and C-S). The catalysts were also recyclable and robust, which are desirable for practical implementation. LnNi-PHI demonstrated impressive substrate scopes that is compatible with electron deficient aryl iodide or low nucleophilic coupling partners. In addition, late-stage diversification of bioactive molecules and pharmaceuticals was successfully accomplished. The design concept/approach of LnNi on PHI is novel, offering exceptional catalytic performance. The reviewer believe this paper would be appeal to broad research communities.

Comment: Thank you very much for reviewing our manuscript. We appreciate the positive comments and constructive suggestions. The manuscript has been carefully revised based on your comments and suggestions.

Specific comments.

1. The surface areas and porosities of PHI before and after LnNi exchange would be helpful.

Comment: Thank you for this valuable suggestion. To investigate the surface area and porosities of PHI before and after LnNi exchange, N₂ adsorption at 77 K was performed. The BET surface areas calculated for the K-PHI and bpyNi-PHI are 25.8 and 41.3 m² g⁻¹ with total pore diameters of 10.3 and 9.94 nm, respectively. An increase in the BET surface area for bpyNi-PHI rationalizes the observed enhanced photocatalytic activity. The corresponding results (Figure S3 and Table S5) were also added to the revised SI as follows:

Figure S13. N₂ adsorption/desorption isotherm (a) and BJH pore size distribution from the N₂ adsorption branch (b) of K-PHI and bpyNi-PHI at 77 K.

Table S5. BET specific surface area, pore diameter and total pore volume.

Sample	Specific surface area (m ² g ⁻¹)	Pore diameter (nm)	Total pore volume (ccg ⁻¹)
K-PHI	25.8	10.3	0.066
bpyNi-PHI	41.3	9.94	0.103

2. EXAFS gave a coordination number of six. However a four-coordinate Ni was shown (from DFT calculation?). Could DFT locate a stable six-coordinate Ni? For example, with solvent or aqua ligand as the author suggested.

Comment: Thank you for pointing this out. We performed the DFT calculation with 2 aqua ligands (bpy-Ni²⁺(H₂O)₂PHI²⁻) and the optimized structure is shown in Figure S16 in Supplementary Information.

3. Could the author provide more discussion on the exceptional activity of LnNi-PHI? For example, a higher loading of Ni on PHI than those on other carbon nitride, or the difference in electronic structure/intrinsic activity of Ni (oxidation states and coordination sphere/bond lengths)

Comment: Thank you for this valuable suggestion. It is noteworthy that the negative PHI 2D-layers play a pivotal role on the enhancement of LnNi²⁺ incorporation and preventing the LnNi leaching due to the strong ion pair interaction. When using other non-metal doped carbon nitride materials such as g-C₃N₄, mpg-CN and CCN instead of K-PHI to prepare LnNi-CN, only minimal incorporation of Ni (0.05–0.23 wt % based on ICP-OES results) was observed. We also examined the catalytic activity of these catalysts including bpyNi-g-C₃N₄, bpyNi-mpg-CN and bpyNi-CCN in C–P cross-coupling reactions. As shown in the following Table, LnNi-CN catalysts exhibited

much lower catalytic activities. We have included these results in Figure 4a with related discussions in the main text. “In addition, other LnNi-CN catalysts exhibited much lower catalytic activities, affording coupling product **1** in low yields, due to the limited Ni loading in these catalysts (**Fig. 4a**, entries 8-10)...”

Entry	catalyst	Yield of 1 (%)
1	bpyNi-PHI (6.0 wt % Nibpy)	92
2	bpyNi-g-C ₃ N ₄ (0.05 wt % Ni)	trace
3	bpyNi-mpg-CN (0.23 wt % Ni)	trace
4	bpyNi-CCN (0.21 wt % Ni)	trace

4. What about the activity comparison between LnNi-PHI and integrated Ni-mpg-CN (reported by Reisner’s group, not Ni-PHI in the manuscript) under the same condition?

Comment: We prepared Ni-mpg-CN according to the reported method by Reisner’s group and examined the catalytic reactivity of Ni-mpg-CN in C–P, C–N, C–O and C–S cross-coupling reactions. Similar to Ni-PHI, Ni-mpg-CN also exhibited much lower activity than LnNi-PHI, affording the corresponding coupling products in low yields (**1**: 21%, **2**: 18%, **3**: 55%, **4**: NR, and **5**: 15%). These results verify the bidentate nitrogen ligands (bpy) play an essential role in dictating the excellent activity.

5. The conversion/yield obtained using semiheterogenous is PHI/NiLn was rather low; however the reported analogue systems exhibit much better performance. Could the authors comment on this?

Comment: The reported semiheterogenous carbon nitride/nickel dual catalysis system used much higher LnNi loading (ca. 5-10 mol%), thus exhibiting satisfied performance. In our system, we tested the semiheterogenous PHI/NiLn system with lower LnNi loading (the same LnNi loading compared to our LnNi-PHI system, cat. 0.07-0.3 mol%), thus giving much lower yield. The reason as to why our LnNi-PHI system can give higher yield is the proximity and cooperativity of the LnNi²⁺ active species and the PHI photocatalyst carrier in LnNi-PHI might facilitate SET and free radical transfer (Table S7–S11).

6. The intact of C-Br during reaction is somehow surprising. Please comment

Comment: We have performed five different C-X bond formation catalytic reactions with aryl bromides instead of aryl iodides (**1**: 40%, **28**: 52%, **3**: 45%, **4**: NR, and **5**: trace). The results indicated that the aryl bromide is suitable for some C-X bond

formation reactions with lower reactivity. We believe the oxidative addition step may be the rate limiting step and the environment on Ni center in LnNi-PHI may lead to more difficult oxidative addition. We have included these results with related discussions in the main text (Figure 6). “Additionally, strong electron deficient aryl bromide is also suitable for C-P, C-O and C-N couplings with lower reactivity (**1**, **3**, **28**)....”

7. It is not clear why LnNi-PHI works better in coupling with primary amine than that with secondary, which is contrary to observations made for other Ni/photoredox catalysts.

Comment: In our catalytic systems, we found that the hydrodehalogenation and hydroxylation of aryl halides were dominant (>60%) in the reaction of aryl iodides with secondary amines, resulting bpyNi-PHI works better in coupling with primary amines than secondary amines. The corresponding results were displayed as follows:

Entry	Yield of 52	Yield of by-product a	Yield of by-product b
1	32%	32%	36%

Entry	Yield of 52	Yield of by-product a	Yield of by-product b
1	31%	33%	36%

We thank the reviewer for the careful evaluation of our manuscript and the constructive suggestions clearly improved our manuscript.

Response to Reviewer #2:

This manuscript reports the use of Ni-functionalized Poly(Heptazine Imide) materials as photocatalysts in a variety of C–P, C–S, C–O, and C–N cross-coupling reactions. The materials are carefully characterized, and the catalytic study comprehensively demonstrate the usefulness of the system reported. Therefore, I recommend its publication after consideration of the following minor aspects:

Comment: Thank you very much for reviewing our manuscript. We appreciate the positive comments and constructive suggestions. The manuscript has been carefully revised based on your comments and suggestions.

1. Figure 1 seems too large.

Comment: Thank you for pointing this out. We have redrawn Figure 1 in the revised manuscript.

2. *1s N XPS of the materials could be informative specially if compared with the pristine PHI material (Which is not presented).*

Comment: The content of N in bipyridine is very low due to the limited bpyNi loading (6.0 wt %), which is difficult to obtain other useful information by analyzing the N 1s XPS spectrum of LnNi-PHI compared to that of K-PHI. The XPS studies of K-PHI results were also added in the revised SI (Figure S6).

3. *Ni-PHI is mentioned in the main text but to understand its nature the reader should go to supplementary information. A very brief explanation in the main text would be appreciated.*

Comment: We have added related information in the main text. “Only trace amounts or low yields of coupled products were observed with Ni-PHI (3.2 wt % Ni, prepared from NiCl₂ and K-PHI without additional bpy ligand)...”

4. *In order to discard homogeneous catalytic processes, an additional experiment is necessary: After one catalytic run, the material is separated by filtration and the formation of products under the reaction conditions is discarded in the absence of material.*

Comment: Thank you for this valuable suggestion. We collected the supernatants of bpyNi-PHI after photoreaction, which should contain the dissolved Ni species. The supernatants then were readded to the sealed tube equipped with starting materials and K-PHI. However, very low conversion (<5%) of the aryl iodide was observed in the C-P coupling reaction.

5. *Some comments on the catalytic mechanism would enrich the manuscript.*

Comment: We added the mechanism discussion in the main manuscript: “Upon the visible light irradiation, the excited catalyst undergoes inner sphere ligand-to-metal charge transfer (LMCT) process to generate the reactive Ni(I) species, thus significantly increasing the catalytic activity over the dual catalytic system that proceeds via outer sphere single electron transfer (SET) between the photocatalyst and nickel catalyst. The resulting Ni(I) species then undergoes oxidative addition with aryl iodide to afford a Ni(III) intermediate, followed by ligand exchange with different types of nucleophiles. Facile reductive elimination at the Ni(III) intermediate delivers the C-heteroatom bond

cross-coupling products along with the regeneration of the active Ni(I) species. To be noted, to prevent the thermodynamically favored comproportionation between Ni(I) and Ni(III) species, continuous blue LED irradiation is required to maintain high concentration of Ni(I) species. (Fig. 1e).”

6. The description of the experimental set-up in supplementary information should include not only the wavelength of light source (Blue LEDs) but also its intensity. Also, how constant temperature is granted in the photocatalytic experiment?

Comment: Thank you for pointing these out. Irradiance of the LED modules was measured using CEL-NP2000 Optical Power and Energy Meter equipped. The output power at 3 cm distance from the light source 19 mW/cm². In order to ensure that the reactions are run near indicated temperature, a simple cooling fan was used to aid in dissipating the heat generated from high power LEDs (Figure S19). We have added these descriptions of the experimental set-up to the revised SI.

We thank the reviewer for the careful evaluation of our manuscript and the constructive suggestions clearly improved our manuscript.

REVIEWERS' COMMENTS

Reviewer #1 (Remarks to the Author):

Previous concerns have been properly addressed.

Reviewer #2 (Remarks to the Author):

After the changes made, this manuscript is suitable for publication in its current form.

Manuscript ID: NCOMMS-22-52573

Response to Reviewer #1:

Previous concerns have been properly addressed.

Comment: Thank you very much for reviewing our manuscript.

Reviewer #2 (Remarks to the Author):

After the changes made, this manuscript is suitable for publication in its current form.

Comment: Thank you very much for reviewing our manuscript.